# Recovery of Adrenal Insufficiency Is Frequent After Adjuvant Mitotane Therapy in Patients with Adrenocortical Carcinoma

**DOI:** 10.3390/cancers12030639

**Published:** 2020-03-10

**Authors:** Jonathan Poirier, Nadia Gagnon, Massimo Terzolo, Soraya Puglisi, Nada El Ghorayeb, Anna Calabrese, André Lacroix, Isabelle Bourdeau

**Affiliations:** 1Division of Endocrinology, Department of Medicine and Research Center, Centre hospitalier de l’Université de Montréal (CRCHUM), Montreal, QC H2X 3E4, Canada; jonathan.poirier.2@umontreal.ca (J.P.); nadia.gagnon@chudequebec.ca (N.G.); ghorayebnada85@gmail.com (N.E.G.); andre.lacroix@umontreal.ca (A.L.); 2Internal Medicine, Department of Clinical and Biological Sciences, San Luigi Hospital, University of Turin, 10043 Orbassano, Italy; terzolo@usa.net (M.T.); sorayapuglisi@yahoo.it (S.P.); anna.calabrese678@gmail.com (A.C.)

**Keywords:** adrenocortical carcinoma, mitotane, HPA axis

## Abstract

Mitotane is a steroidogenesis inhibitor and adrenolytic drug used for treatment of adrenocortical cancer (ACC). Mitotane therapy causes adrenal insufficiency requiring glucocorticoid replacement in all patients. However, it is unclear whether chronic therapy with mitotane induces complete destruction of zona fasciculata and whether hypothalamic-pituitary-adrenal (HPA) axis can recover after treatment cessation. Our objective was to assess the HPA axis recovery in a cohort of patients after cessation of adjuvant mitotane therapy for ACC. We retrospectively reviewed patient files with stage I-II-III ACC in two referral centers in Canada and Italy. Data on demographics, tumor characteristics, hormonal profile, and HPA axis were collected. Data from 23 patients with pathologically proven ACC treated with adjuvant mitotane for a minimum of two years were analyzed. Eight patients were males and 15 were females and the median age was 41 years old (range 18 to 73). After mitotane cessation, 18/23 (78.3%) patients achieved a complete HPA axis recovery while 3/23 (13.0%) were unable to tolerate glucocorticoid withdrawal despite having normal hormonal test values and 2/23 (8.7%) never achieved recovery. The mean time interval between mitotane cessation and HPA axis recovery was 2.7 years. A high proportion of patients achieved HPA axis recovery following cessation of mitotane adjuvant therapy. However, complete recovery was often delayed up to 2.5 years and regular assessment of the hormonal profile is required.

## 1. Introduction

Adrenocortical carcinoma (ACC) is a rare but often aggressive tumor with an incidence of one to two cases per million per year. More than half of cases are diagnosed at an advanced stage and present with a low five-year disease-free survival [1,2]. In previous studies, adjuvant treatment with mitotane was shown to increase recurrence-free survival in patients with radically resected ACC [3,4,5]. Surgically removable tumors of stages I to III presenting with evidence of loco-regional invasion or aggressive pathologic features such as a high proliferation index (Ki67 > 10%) or a positive margin status are candidates for this therapy [6,7,8]. This is highlighted by a recent consensus from the European association of endocrinology and European Network for the Study of Adrenal Tumors (ENSAT), which recommends adjuvant mitotane at therapeutic levels for at least two years in patients operated for ACC with high recurrence risk [1].

Mitotane is an analogue of dichloro-diphenyl-trichloroethane (DDT) insecticide and has been used for ACC since the 1960s. It is known to cause frequent side effects including gastro-intestinal and hepatic disturbances (anorexia, nausea, vomiting and diarrhea, hepatitis, or liver enzyme elevation), central nervous system toxicity (lethargy, somnolence, dizziness, and vertigo), leukopenia, increased serum cholesterol, altered drug metabolism, and endocrine disorders such as hypogonadism and hypothyroidism [9,10]. Mitotane has significant adrenolytic properties, which causes adrenal insufficiency and, for this reason, an actual standard of care requires that simultaneous replacement with sufficient glucocorticoid be given to patients receiving mitotane. Less frequently, mineralocorticoid replacement is also required [1]. In vitro studies showed that mitotane causes interference with cholesterol metabolism and with mitochondrial activity in adrenocortical cells, which often leads to an increase in the apoptosis rate. It also reduces adrenal steroidogenesis by inhibiting key mitochondrial enzymes like CYP11B, which reduces the conversion of hormone precursors into active hormones such as cortisol [11,12,13,14]. It was also found that mitotane could potentially interfere with pituitary function by comparing adrenocorticotropic hormone (ACTH) levels between patients treated with mitotane for ACC and patients with primary adrenal insufficiency (autoimmune or post-adrenalectomy) [15]. Basal values and post-CRH stimulation test (100 mcg) both showed a statistically significant decrease in ACTH levels in patients treated with mitotane, which suggests that this drug might have inhibitory effects at diverse levels of the hypothalamic-pituitary-adrenal (HPA) axis. Data on whether adjuvant mitotane causes permanent adrenal insufficiency or recovery of the HPA axis can occur after mitotane withdrawal remains limited [16]. We present data from two referral centers for ACC patients in Canada and Italy indicating that the HPA axis recovery can occur in a high proportion of patients several months after cessation of mitotane therapy.

## 2. Results

### 2.1. Patient Profiles

A total of 48 patients with a pathologically proven ACC were identified to have minimally completed a two-year course of adjuvant mitotane therapy (34 and 14 patients from Italian and Canadian centers, respectively). Among those, 23 patients (12 from Italy and 11 from Canada) had sufficient data to interpret the HPA axis recovery status. In this group, all were Caucasians except for one patient with middle Eastern origin. Eight patients were males and fifteen were females. The median age was 41 years old (range 18 to 73). Table 1 summarizes data of the studied cohort of patients.

Five out of 23 patients (21.7%) were stage I, 14/23 (60.9%) were stage II, and 4/23 (17.4%) were stage III with a mean ACC size of 10 cm (range 3 to 26 cm). The hormonal secretion profile was available before the ACC surgical resection in 14 out of 23 patients: 6 were non-secreting, 5 were producing cortisol only, two had an androgen-cortisol co-secretion, and one was producing androgen only. Negative margin (R0) was achieved in 14/23 (60.9%) patients and 12 of the tumours had a high proliferation index (Ki67 > 10%) (range 2 to 67). Surgical laparoscopy was performed in seven patients that had an ACC size < 8 cm except for one tumour with a size of 11 cm. Surgical laparotomy was performed for every other tumour (except for two with unknown procedures). Only 1/23 patients had a metastatic recurrence of ACC eight years after the initial diagnosis.

### 2.2. Mitotane Exposure

All 23 patients had a 24-month minimal exposure to mitotane therapy with a median duration of 34 months (range from 24 to 85 months). Mitotane therapeutic level duration (mitotane blood level > 14 mg/mL) was shorter than 24 months in 12 patients and could not be determined in three patients because mitotane blood levels were not available. Mitotane mean maximal blood concentration was 19.4 mg/mL, which range from 8.1 mg/mL to 27.1 mg/mL, with two patients never achieving a maximal concentration above 14 mg/mL. Maximal daily dose varied from 2 g/day to 4.5 g/day in studied patients. Only two patients had to discontinue their treatment after 24 and 31 months, respectively, secondary to side effects.

### 2.3. Glucocorticoid Supplementation 

All patients received a glucocorticoid supplementation with either hydrocortisone (12 patients) or cortisone acetate (10 patients) during and after mitotane therapy (one patient with unknown treatment). Mineralocorticoid replacement with fludrocortisone was also necessary for eight patients (34.8%). The mean maximal daily replacement dose was 69.4 mg (range from 50 to 87.5 mg) for cortisone acetate and 51.6 mg (range from 25 to 100 mg) for hydrocortisone during mitotane therapy.

### 2.4. HPA Axis Recovery

A total of 18/23 patients (78.3%) achieved a complete recovery of HPA after mitotane cessation. Of the five remaining patients, 3/23 (13.0%) achieved a partial recovery with normal laboratory test values but clinical inability to taper glucocorticoid replacement due to symptoms of adrenal insufficiency. Lastly, 2/23 patients (8.7%) never achieved recovery of their HPA axis despite having stopped mitotane therapy for more than two and six years, respectively. The mean interval of time between the mitotane’s last dose and corticosteroid replacement cessation was 999.6 days (2.7 years) and the median was 1012 days (2.8 years) (range from 106 to 2991 days) (Figure 1). Eight patients received simultaneous mineralocorticoid supplementation and fludrocortisone replacement was stopped at the time of corticosteroid cessation in all except one patient who failed to achieve HPA axis recovery altogether.

## 3. Discussion

In this cohort of patients with ACC evaluated for HPA axis recovery status after cessation of mitotane therapy, we found that approximately three out of four patients eventually had a return of a normal HPA axis. This finding suggests that residual adrenocortical tissue remains viable and functional despite prolonged exposure to mitotane and its known adrenolytic effect on adrenocortical cells [11].

Recovery was a very slow process and occurred within a span of several months to many years in most patients. A possible culprit for this latency was initially suspected to be mitotane’s long half-life and lipophilic properties that allows it to accumulate in fat tissue [12]. However, this hypothesis is unlikely, considering that almost every patient had undetectable blood levels of mitotane at the time of hormonal evaluation, including subjects without HPA axis recovery. Patients with an extended mitotane exposure (up to 3.5 years) followed a similar evolution course to those with shorter treatment exposure including those with therapeutic duration of less than two years. In previous studies, there has been conflicting evidence regarding potential pharmacologic activity of mitotane’s metabolites, but the most recent data tends to dismiss the relevance of metabolites’ activity in ACC treatment and monitoring [17,18,19]. For this reason, metabolites’ measurement is not routinely performed in medical laboratories and is not expected to have any impact on adrenal function [20].

Other potential variables were explored in search of a cause for the absence or delayed recovery of HPA axis recovery. Boulate et al. recently demonstrated a potentiating effect of rosuvastatin on mitotane cell toxicity in vitro [21]. Van Koetsveld et al. showed that mitotane could have an increased in vitro sensitivity on cortisol-producing ACC tumors [13]. Cusato et al. hypothesized that mitotane blood levels and dosage could be affected by body fat and seasonal variations in white and brown adipose tissue [22]. Data revision with attention given to variables such as statin drug use, initial tumor characteristics (size, hormonal profile, proliferation index), surgery type, and outcome did not reveal any specific relation with an HPA axis recovery status. We observed, however, that two of the five patients with absent or partial recovery were the patients with the highest body weight in the Canadian cohort with 108.9 kg and 130 kg, respectively (third highest was 91.5 kg). Due to the low number of patients, the body weight impact on HPA axis recovery could be a coincidence in this cohort and this association needs to be further explored. Five patients in our cohort had Cushing’s syndrome at presentation with consequent HPA axis suppression. They all achieved a relatively rapid (3.5 to 26 months) and complete HPA axis recovery following mitotane withdrawal with normal or high ACTH values.

Reduction of pituitary ACTH secretion by mitotane has been described previously as a potential etiology for HPA axis suppression. Reimondo et al. found that mitotane-treated patients with adrenal insufficiency had lower ACTH levels than patients with primary adrenal insufficiency of other etiologies [15]. In our cohort, this finding was also present but not consistent among patients. An inappropriately low ACTH level was observed in only two of the five patients with abnormal HPA axis recovery.

In 10 patients, ACTH stimulation testing was not performed and HPA axis recovery was determined based on a maximal morning cortisol value. In these cases, partial adrenal insufficiency may not be entirely ruled out [23]. However, none of these patients developed symptoms of adrenal insufficiency during at least one year follow up after cessation of hydrocortisone replacement.

Overall, follow-up of ACC patients following mitotane cessation was relatively uneventful. At the time of data analysis, only 1 of the 23 patients presented a metastatic recurrence of ACC eight years after the initial surgery and was among the two patients unable to stop glucocorticoid replacement despite having a normal HPA axis recovery with a morning cortisol value of 425 nmol/L. Since no patient achieved a complete HPA axis recovery and presented signs of recurrent ACC disease, concerns about a possible lack of mitotane cytotoxic effect on adrenal tumor cells in this setting seems unlikely. However, the low rate of recurrence could also be explained by the higher prevalence of stage I–II tumors compared to stage III tumors in this cohort of patients, which naturally presents as a less advanced disease and has a better overall survival. In addition, these patients were selected on their capacity to discontinue adjuvant mitotane after a minimal period of two years after surgery, which means they were disease-free.

One of the drawbacks of our study is its retrospective design. Despite being reviewed thoroughly, patient files were sometimes incomplete. This could potentially cause underestimation or overestimation of some results due to missing data. Another weakness resides in the low number of patients, which makes it difficult to validate previous hypothesis. Because of this, many findings presented are exploratory and would need further confirmation in larger cohorts of patients.

This is, to our knowledge, the first study directly addressing the HPA axis recovery in vivo after mitotane cessation in the patient with ACC. The collaborative nature of this study including data from a Canadian center and an Italian center was useful given the rarity of ACC.

## 4. Materials and Methods 

Patients’ data was retrieved in a retrospective manner from electronic and paper charts in two large referral centers: (1) Centre hospitalier de l’Université de Montréal (CHUM), Canada and (2) San Luigi Hospital, Orbassano, University of Turin, Italy. After obtaining institutional ethical committee approval from each center, CE2010-304, 09.244 (CHUM), and CE 36/2012 (San Luigi Hospital), every case from 1992 (Italy) and 1995 (Montreal) to this day with pathologically confirmed ACC of stage I-II-III based on ENSAT classification [24] was reviewed. Data regarding patient demography, tumour size, hormonal secretion, type of surgery, duration of adjuvant mitotane therapy, mitotane serum levels, and corticosteroid replacement was collected and analyzed.

Patients who had completed at least two years of adjuvant mitotane therapy and stopped the treatment were evaluated for recovery of the HPA axis with a morning cortisol level (before glucocorticoid supplementation), morning ACTH level (normal range 2.0–11.0 pmol/L in Montreal and 1.32–13.2 pmol/L in Italy), and 1 or 250 mcg ACTH 1–24 stimulation test if applicable. For the ACTH stimulation test, patients were asked to stop taking their glucocorticoid supplementation after the previous morning dose. Cortisol blood levels were measured before intravenous ACTH 1–24 (Cortrosyn, Organon Canada) administration (T0) after 30 min (T30) and 60 min (T60) following the injection. Mitotane blood concentration was determined by a High-Pressure Liquid Chromatography method (HPLC) in both centers. Mitotane blood levels were considered therapeutic when a target of 14–20 mg/mL was reached. Cortisol determinations were performed using a chemiluminescence immunoassay in both centers (Montreal: ADVIA Centaur, Siemens Healthcare Diagnostics Ltd, Mississauga, ON, intra and inter-assay CV varying from 3% to 3.8% and 1.9% to 5.5%, respectively, and Turin: CLIA, Siemens, intra and inter-assay variation coefficient of 3.7–4.2% and 4.4–6.0%, respectively).

Following interpretation of laboratory values, the HPA axis recovery status was determined. In the Canadian center, reference values for the HPA axis evaluation were considered normal if AM serum cortisol was above 280 nmol/L or if serum cortisol after ACTH stimulation test reached a peak level of at least 450 nmol/L. In the Italian center, normal reference values were above 331 nmol/L for an AM serum cortisol value or above 552 nmol/L for the peak cortisol level during the ACTH stimulation test. Patients were classified as having either (1) complete HPA axis recovery, (2) normal test values but inability to withdraw from glucocorticoid supplementation, or (3) insufficient HPA axis recovery.

## 5. Conclusions

Our results show that 78% of patients treated with adjuvant mitotane therapy after ACC resection achieve complete HPA axis recovery and are able to discontinue glucocorticoid replacement. This favorable outcome generally occurs late after mitotane cessation with most patients being glucocorticoid supplemented for more than two years after the end of mitotane therapy and reaffirms the necessity for follow-up to determine the need for continuous hormonal replacement in these patients. Unfortunately, we did not highlight any predictors for HPA axis recovery after mitotane exposure. Further studies are needed with larger cohorts of patients to give more insights on this clinical challenge.

## Figures and Tables

**Figure 1 cancers-12-00639-f001:**
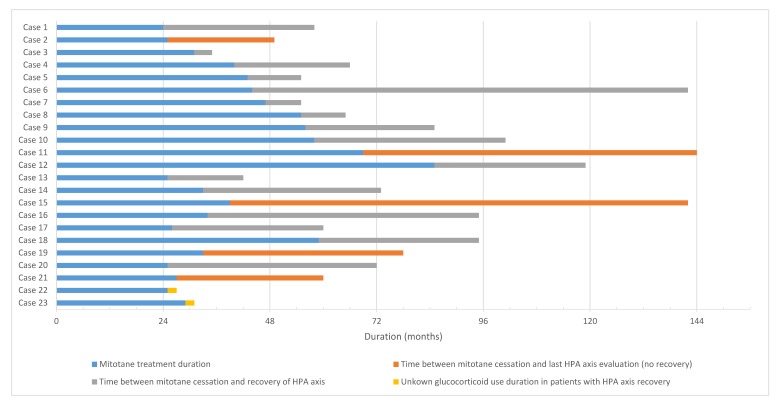
Schematic representation of mitotane exposure and HPA axis recovery timeline for each patient.

**Table 1 cancers-12-00639-t001:** Summarized data of each patient with adrenocortical cancer (ACC) treated with mitotane including tumor characteristics, treatment, and HPA axis evaluation.

ID	Study Center	Sex	Age at Diagnosis	Hormonal Secretion	ENSAT Stage	Tumor Size	Mitotane Duration	Mitotane Peak Level	Type of Steroid Replacement	Maximal Dose of Glucocorticoid	Highest Morning Cortisol (8 h)	Morning ACTH at Recovery (8 h)	Maximal Cortisol Peak during ACTH Test
No			(Years)			(cm)	(Months)	(mg/mL)		(HC Equivalent mg)	(nmol/L)	(pmol/L)	(nmol/L)
1	Italy	F	29	None	I	4	24	25.0	CA	60	408	40.3	NA
2 *	Italy	F	32	NA	I	5	25	15.0	CA	40	229	8.8	NA
3	Italy	F	55	C	II	14	31	17.0	CA	65	339	205.0	NA
4	Italy	F	59	C	II	10	40	20.0	CA and F	60	373	72.2	NA
5	Italy	F	48	NA	II	16	43	17.8	CA and F	60	505	13.2	637.3 (a)
6	Italy	F	29	A	II	13	44	19.0	CA and F	40	337	46.2	350.4 (b)
7	Italy	M	27	C	II	9	47	20.0	CA	60	486	12.0	NA
8	Italy	F	48	C	I	3	55	20.0	CA	70	353	23.1	NA
9	Italy	F	45	None	II	9	56	20.0	HC	25	301	14.1	582.2 (a)
10	Italy	M	18	NA	II	17	58	28.0	CA and F	50	422	80.3	482.8 (a)
11 *	Italy	M	54	NA	II	11	69	25.0	HC	60	279	120.0	309 (a)
12	Italy	F	55	NA	II	13	85	24.0	CA	50	475	11.2	NA
13	Canada	F	35	C	I	5	25	21.1	HC	40	391	8.8	586 (a)
14	Canada	M	53	None	I	3	33	22.7	HC and F	50	NA	NA	594 (a)
15 †	Canada	F	39	NA	II	7	39	25.5	HC	60	NA	NA	454 (b)
16	Canada	F	21	NA	II	7	34	17.6	HC and F	35	122	23.5	605 (a)
17	Canada	F	51	None	II	13	26	27.1	HC	60	466	3.3	477 (b)
18	Canada	M	73	NA	III	5	59	11.9	HC	30	374	12.2	538 (b)
19 †	Canada	F	31	None	II	21	33	20.1	HC and F	60	425	71.7	321 (b)
20	Canada	M	37	None	II	10	25	8.1	HC	45	65	5.8	512 (a)
21 †	Canada	F	41	A and C	III	12	27	20.3	HC	55	310	8.1	NA
22	Canada	M	31	NA	III	26	25	18.0	NA	NA	473	7.9	NA
23	Canada	M	62	A and C	III	5	29	19.0	HC and F	NA	432	2.2	NA

* no HPA axis recovery, †failure at glucocorticoid withdrawal attempt, value corresponding to the highest morning cortisol, M: male, F: female, C: cortisol, A: androgen, ENSAT: European Network for the Study of Adrenal Tumors, CA: cortisone acetate, HC: hydrocortisone, F: fludrocortisone, NA: not available, ACTH: adrenocorticotropic hormone, a: 250-mcg dose for ACTH stimulation test, b: 1-mcg dose for ACTH stimulation test.

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
