# Peer review of "Recovery of Adrenal Insufficiency Is Frequent After Adjuvant Mitotane Therapy in Patients with Adrenocortical Carcinoma"

_cancers, 2020, doi:10.3390/cancers12030639_

Round 1
Reviewer 1 Report
The Authors presented in a clear way analysis of recovery of HPA axis function in ACC patients after mitotane cessation. The results are interesting and broadens the knowledge about mitotane treatment. The main limitation of the study is the small sample size, but it is associated with the rarity of ACC.
I think that potentiating effect of statins on mitotane cell toxicity is noteworthy. I suggest that the authors could check who was treated with statins and analyze the effect of statin treatment on the restoration of the HPA axis function in the study group.
Author Response
Comment 1: I think that potentiating effect of statins on mitotane cell toxicity is noteworthy. I suggest that the authors could check who was treated with statins and analyze the effect of statin treatment on the restoration of the HPA axis function in the study group.
Response to reviewer:
We analyzed the HPA axis recovery patterns in patients treated with a statin and those who did not. Among the Canadian cases, 4 patients received statins (either rosuvastatin 5-10mg or atorvastatin 10mg). In the Italian group of patients, 2 had a moderate dose of statin (atorvastatin 20mg) and 1 was taking monacolin k extract, a nutraceutical compound similar to lovastatin extracted from red yeast rice. Two of the patients with low dose statin were unable to stop hydrocortisone replacement despite having normal cortisol laboratory tests. The remaining 5 patients had a delay of recovery comparable to the rest of the cohort. Due to the low number of cases in our cohort, it is difficult to draw conclusions on the impact of statin use on HPA axis recovery.
Reviewer 2 Report
This is an interesting and clinically relevant retrospective study on HPA axis recovery following mitotane treatment for adrenocortical carcinoma. The results are provided in a clear fashion and the manuscript is well written. Please find below some comments/suggestions:
1. The authors state that sufficient data to evaluate HPA axis were only available in 23 out of 48 patients. What percentage of the remaining 25 patients continue glucocorticoid replacement therapy?
2. Are factors such as the cumulative mitotane dose administered to each patient, and the “therapeutic duration” (defined as mitotane blood level 87 >14mg/mL) known? Do these factors correlate with the risk of developing adrenal insufficiency or the time to recovery of HPA axis? Were patients who did not achieve optimal therapeutic mitotane blood levels less likely to develop adrenal insufficiency?
3. Although ACTH stimulation was performed in the majority of the cases included in the study, HPA axis evaluation was based on morning cortisol values in 10 patients. However, some studies suggest that a morning serum cortisol alone may not be a reliable indicator for the diagnosis or exclusion of adrenal insufficiency (Erturk et al, J Clin Endocrinol Metab 1998 83 (7), 2350-4). This may be relevant to the interpretation of HPA axis recovery in some of the cases included in the study. In particular: Case 6 had a subnormal response on ACTH stimulation testing yet was considered as having an intact HPA axis based on a morning cortisol of 336 nmol/L. Case 2 was considered as having adrenal insufficiency based on a morning cortisol of 229 nmol/L alone. Cases 3, 4 and 8 are considered as having an intact HPA axis based on a morning cortisol of 339, 372 and 353 nmol/L respectively; however, as no ACTH stimulation testing results are available, partial adrenal insufficiency may not be entirely ruled out.
4. Table 1. Please define “ACTH at recovery”? Is this the value that corresponds to the “highest morning cortisol result”?
5. Line 17: change “fasciculate” to “fasciculata”
Author Response
Comment 1: The authors state that sufficient data to evaluate HPA axis were only available in 23 out of 48 patients. What percentage of the remaining 25 patients continue glucocorticoid replacement therapy?
Response to reviewer:
Data files of the remaining 25 patients were incomplete and/or had a follow-up duration of less than 2 years after Mitotane cessation. Considering that the mean time for HPA axis recovery was 2.7 years in our cohort, we think that their follow-up duration was insufficient to properly interpret their HPA axis recovery status. As stated on line 176 in our discussion, we are aware that this may have lead to under or overestimation of frequency of HPA axis recovery
"Despite being reviewed thoroughly, patient files were sometimes incomplete. This could potentially cause underestimation or overestimation of some results due to missing data.”.
Comment 2: Are factors such as the cumulative mitotane dose administered to each patient, and the “therapeutic duration” (defined as mitotane blood level 87 >14mg/mL) known? Do these factors correlate with the risk of developing adrenal insufficiency or the time to recovery of HPA axis? Were patients who did not achieve optimal therapeutic mitotane blood levels less likely to develop adrenal insufficiency?
Response to reviewer:
The cumulative dose of mitotane was not determined in our patients. Therapeutic mitotane duration was known in 20/23 patients but there was no correlation with the risk of adrenal insufficiency or timing of HPA axis recovery (line 87 and 131). Only 2 out of 23 patients did not achieve therapeutic mitotane blood level and both patients achieved complete recovery of their HPA axis (case 18 and 20 in Table 1). Delay for recovery in these patients was of 34 and 47 months respectively. Also, during revision of therapeutic mitotane duration in the cohort from Italy, 6 patients were incorrectly classified as having reached a therapeutic exposure > 24 months. However, we found no difference in our interpretation of data after considering this change. A correction was made on line 89.
Comment 3: Although ACTH stimulation was performed in the majority of the cases included in the study, HPA axis evaluation was based on morning cortisol values in 10 patients. However, some studies suggest that a morning serum cortisol alone may not be a reliable indicator for the diagnosis or exclusion of adrenal insufficiency (Erturk et al, J Clin Endocrinol Metab 1998 83 (7), 2350-4). This may be relevant to the interpretation of HPA axis recovery in some of the cases included in the study. In particular: Case 6 had a subnormal response on ACTH stimulation testing yet was considered as having an intact HPA axis based on a morning cortisol of 336 nmol/L. Case 2 was considered as having adrenal insufficiency based on a morning cortisol of 229 nmol/L alone. Cases 3, 4 and 8 are considered as having an intact HPA axis based on a morning cortisol of 339, 372 and 353 nmol/L respectively; however, as no ACTH stimulation testing results are available, partial adrenal insufficiency may not be entirely ruled out.
Response to reviewer:
We thank the reviewer for this important comment. We now discuss this point in the manuscript line 158:”
In 10 patients, ACTH stimulation testing was not performed and HPA axis recovery was determined based on maximal morning cortisol value. In these cases, partial adrenal insufficiency may not be entirely ruled out. However, none of these patients developed symptoms of adrenal insufficiency during at least one year follow up after cessation of hydrocortisone replacement.
.” We also added the suggested reference on line 160 (ref 23).
Comment 4: Table 1. Please define “ACTH at recovery”? Is this the value that corresponds to the “highest morning cortisol result”?
Response to reviewer:
It is the ACTH value corresponding with the highest morning cortisol. We added the symbol “ ¶ ” with the appropriate definition in the legend of Table 1.
Comment 5: Line 17: change “fasciculate” to “fasciculata”
Response to reviewer: Thank you for this suggestion. The change was made.
Reviewer 3 Report
The paper “Hypothalamic-pituitary-adrenal axis recovery after adjuvant mitotane therapy in patients with adrenocortical carcinoma” is a retrospective study aimed to evaluate the HPA axis recovery after cessation of mitotane treatment. The paper demonstrated that a high proportion of patients achieved “HPA axis recovery” following the withdrawal of mitotane therapy.
I have the following comments:
1. The most important point is the definition of “HPA axis recovery”. As well know, adrenolytic mitotane activity causes primary adrenal insufficiency, a condition associated with consensual ACTH increase. Furthermore, mitotane could potentially reduce ACTH levels, causing a potential direct inhibitory effect on HPA axis, as mentioned by the Authors. The title reports "HPA axis recovery", as if the inhibitory mechanism on HPA axis was the main cause of hypoadrenalism. However, in this group of patients, this finding seems to be not consistent among patients, as reported in the discussion section. Therefore, the most prevalent mechanism is the primary adrenal insufficiency induced by mitotane directly on adrenal gland. So, is it correct to state "HPA axis recovery"? In my opinion, the title should be modified.
2.A very important point should be emphasized: 5 patients had adrenal Cushing syndrome. In these cases, a "tertiary" component of hypoadrenalism is certainly expected. In my opinion, this group of patients should be considered separately.
3.Considering the heterogeneous group of patients, it may be useful to add in the table the follow-up data (including the trend of ACTH and renin values during treatment with mitotane) and the test used to test the recovery of adrenal function (250 or 1 mcg of ACTH 1-24).
4.More recent references about mitotane should be added (e.g. 10.17925/EE.2018.14.2.62).
Author Response
Comment 1: The most important point is the definition of “HPA axis recovery”. As well know, adrenolytic mitotane activity causes primary adrenal insufficiency, a condition associated with consensual ACTH increase. Furthermore, mitotane could potentially reduce ACTH levels, causing a potential direct inhibitory effect on HPA axis, as mentioned by the Authors. The title reports "HPA axis recovery", as if the inhibitory mechanism on HPA axis was the main cause of hypoadrenalism. However, in this group of patients, this finding seems to be not consistent among patients, as reported in the discussion section. Therefore, the most prevalent mechanism is the primary adrenal insufficiency induced by mitotane directly on adrenal gland. So, is it correct to state "HPA axis recovery"? In my opinion, the title should be modified.
Response to reviewer:
We thank the reviewer for his/her comments and we now propose a new title for the manuscript based on his/her comments :
Recovery of adrenal insufficiency is frequent after adjuvant mitotane therapy in patients with adrenocortical carcinoma.
Comment 2: A very important point should be emphasized: 5 patients had adrenal Cushing syndrome. In these cases, a "tertiary" component of hypoadrenalism is certainly expected. In my opinion, this group of patients should be considered separately.
Response to reviewer:
We agree that patients with Cushing’s syndrome secondary to ACC are expected to have a central inhibition of their HPA axis caused by excess of cortisol. However, hypercortisolism was resolved after the surgery and these patients had no recurrence of their tumor. Considering that hormonal assessment was performed more than 2 years after surgical resection of ACC in all patients, it is unlikely that HPA axis suppression persists more than one year after resection. However, it is possible that replacement therapy with high glucocorticoid doses may extend the HPA axis suppression in these patients by a similar mechanism. However, this was not observed in our patients with initial Cushing’s syndrome and interestingly, they all had high ACTH values at the time of adrenal insufficiency recovery (some had the highest values). They also presented with the fastest recovery times in all of the cohort, ranging from 3.5 to 26 months between mitotane cessation and HPA axis recovery. A new point was added to our discussion on line 148.
“Five patients in our cohort had Cushing’s syndrome at presentation with consequent HPA axis suppression. Interestingly, they all achieved a relatively rapid (3.5 to 26 months) and complete HPA axis recovery following mitotane withdrawal with normal or high ACTH values.
Comment 3.Considering the heterogeneous group of patients, it may be useful to add in the table the follow-up data (including the trend of ACTH and renin values during treatment with mitotane) and the test used to test the recovery of adrenal function (250 or 1 mcg of ACTH 1-24).
Response to reviewer:
Unfortunately, trends of ACTH and renin values were not available in most patients. The ACTH dose used in the stimulation test is already indicated in Table 1 with the letter “a” or “b” next to the cortisol value in last column. We increased the size of the indicator to make it more visible.
Comment 4. More recent references about mitotane should be added (e.g. 10.17925/EE.2018.14.2.62).
Response to reviewer:
The reference was added on line 49 as ref 10 with adjustment of all references.